# Towards Pretraining Text Encoders for TabPFN

**Mustafa Tajjar** [* 1 2]  **Alexander Pfefferle** [* 3 1]  **Lennart Purucker** [4 1]  **Frank Hutter** [4 3 1]

## Abstract

Tabular foundation models, such as TabPFN, achieve strong performance on tabular datasets with numerical and categorical data, but do not natively handle high-cardinality text features. Standard pipelines, therefore, embed text with a language model and compress the resulting vectors with PCA into a small number of scalar features before inputting them into TabPFN. This creates an information bottleneck: most embedding dimensions are discarded, and the compressed representation must then be expanded again by TabPFN's feature encoder. End-to-end alternatives can avoid PCA, but they require large amounts of pretraining data containing text cells and usually perform subpar compared to tabular foundation models that were pretrained on large amounts of synthetic data. Inspired by modality-alignment approaches like LLaVA (vision-to-LLM token projection) and TableGPT-style systems (table-to-LLM token projection), we introduce the **TabPFN Text Adapter** (text-to-TFM token projection). We freeze both the sentence encoder and TabPFN, and train only a lightweight adapter that maps text embeddings into a short sequence of tokens in TabPFN's embedding space. This design removes the PCA bottleneck, preserves TabPFN's numerical strengths, and is more efficient to train than end-to-end text-tabular pipelines.

## 1. Introduction

Recent progress in tabular foundation models has markedly improved predictive learning on structured data, with

---

[1]Department of Computer Science, University of Freiburg, Freiburg, Germany [2]Zuse School ELIZA `https://eliza.school/` [3]ELLIS Institute Tübingen, Tübingen, Germany [4]Prior Labs, Freiburg, Germany. Correspondence to: Mustafa Tajjar <tajjarm@cs.uni-freiburg.de>, Alexander Pfefferle <pfeffera@cs.uni-freiburg.de>.

*Proceedings of the $2^{nd}$ ICML Workshop on Foundation Models for Structured Data*, Seoul, South Korea. 2026. Copyright 2026 by the author(s).

TabPFN (Hollmann et al., 2025) as a representative example. However, handling real-world datasets often requires processing mixed signals, including text, categoricals, and numericals. Currently, TabPFN cannot leverage textual data natively; text fields must be treated as categorical or preprocessed via a pipeline consisting of verbalization, language model (LM) embedding, and subsequent downprojection. Typically, this projection relies on principal component analysis (PCA) (F.R.S., 1901) to reduce the high-dimensional LM embeddings into a few numerical features. This PCA step introduces an information bottleneck, stripping away semantic richness before the data even reaches the tabular model. It also prevents the pipeline from being fully differentiable, thus taking away the ability to fine-tune the language model based on TabPFN's predictive signal.

Recent approaches, such as ConTextTab (Spinaci et al., 2025) and TabSTAR (Arazi et al., 2025), attempt to natively align text embeddings with tabular models via end-to-end pretraining and, in TabSTAR's case, fine-tuning per dataset. While they can be effective for text-focused tasks, these end-to-end methods present drawbacks like locking the model to a specific LM and verbalization strategy, and critically, often degrade the model's predictive performance on purely numerical and categorical signals (Erickson et al., 2025), which could be attributed to being predominantly trained on real-data collections, such as T4 (Gardner et al., 2024). Such real-data collections are often of low quality for pretraining purposes due to being mostly web scrapes that do not necessarily represent a predictive task (Garg et al., 2025).

To address this, we investigate a lightweight, post-training alignment strategy inspired by visual instruction tuning for Large Language Models, such as LLaVA (Liu et al., 2023), and tabular instruction tuning like TableGPT (Wang et al., 2023). We introduce the **TabPFN Text Adapter**, which directly projects LM embeddings into the embedding space of TabPFN without updating the weights of either foundation model.

In our approach, textual features are separated from numerical and categorical features. We leave TabPFN to process both numerical and categorical features using its standard encoder to preserve its robust performance. Simultaneously, our adapter pipeline processes the text: a sentence transformer generates a high-dimensional embedding, which is

then normalized and passed through a pretrained MLP. This MLP maps the text representation directly into a small sequence (e.g., 1 to 10 tokens) of TabPFN's 192-dimensional embedding space. These adapted text tokens are concatenated with the standard tabular embeddings and fed into TabPFN's transformer blocks. By pretraining this adapter on a diverse set of datasets with text columns, we provide a simpler, cheaper, and more flexible alternative to end-to-end pretraining, taking a vital first step toward optimizing the full text-encoder pipeline for TabPFN.

## 2. Related Work

**Foundation Models for Tabular Data with Text.** ConText-Tab (Spinaci et al., 2025) natively supports textual features by combining a frozen language model with a lightweight text projector inside a tabular foundation model pipeline. It is pretrained on T4, a large corpus of real-world tables. While ConTextTab is strong on text-rich benchmarks, the TabArena benchmark leaderboard reports that on non-text tabular tasks, it is outperformed by a broad set of competing models (Erickson et al., 2025), suggesting a trade-off between text specialization and general tabular performance. TabSTAR (Arazi et al., 2025) also targets text-aware tabular learning, but it requires task-specific fine-tuning of the text encoder. This improves adaptation to each dataset, yet increases training cost and departs from the pure in-context-learning regime used by TabPFN-style inference. Our adapter design is motivated by multimodal alignment methods in language modeling. LLaVA aligns visual features (from a vision encoder) to the token space of a frozen LLM through a learned projector (Liu et al., 2023). TableGPT similarly maps structured table inputs into an LLM-compatible token space for downstream reasoning and generation. We transfer this alignment principle to tabular prediction by mapping LM text embeddings directly into TabPFN's internal token space while keeping both backbones frozen.

## 3. Method

During the forward pass with a new tabular dataset, we separate columns containing free text from numerical and categorical columns. We use TabPFN 2.5 (Grinsztajn et al., 2026) as the frozen tabular foundation model, keeping all components unchanged. TabPFN embeds numerical and categorical features in its own way: numerical features are standardized along their respective columns, grouped, and passed through a linear layer (classification) or a two-layer MLP (regression), mapping each feature group to a 192-dimensional embedding in a cell-wise manner. We provide an overview of our method in Figure 1.

**Text encoding and adapter architecture.** For free-text columns we first verbalize each cell by prepending the column name, transforming a raw cell value `text` into `column_name:  text`. This simple contextualization helps the language model disambiguate identical text values that carry different meanings depending on their column. The verbalized cell is then encoded with a language model via the sentence-transformer library (Reimers & Gurevych, 2019) — in our case `all-MiniLM-L6-v2`[1], which is based on the MiniLM architecture (Wang et al., 2020) — producing a high-dimensional embedding for each cell. Our adapter then maps from the sentence-transformer embedding space to a small number of tokens in TabPFN's 192-dimensional embedding space. To maintain architectural consistency, the adapter mirrors TabPFN's feature encoder: a linear layer for classification and a 2-layer MLP for regression. Based on our ablation study (Section 5), we use five tokens per text cell, as it is close in performance to the larger 10 tokens for classification, while less computationally expensive, and one token for regression, as it yields the best results. Once all cell embeddings are produced, they are passed through TabPFN's layers.

**Pretraining data.** Since adapter training requires tables with meaningful textual columns, we use a subset of the STRABLE benchmark (Blayer et al., 2026), the specific datasets are listed in Appendix C. To distinguish free-text columns from textual categoricals — the latter of which should go through the standard TabPFN pipeline — we check whether the number of unique values in a column exceeds 20. For each training step, we sample a random dataset and subsample a random set of rows and columns, ensuring at least one text column is included.

**Training procedure.** During training, we freeze the entire sentence transformer and TabPFN architectures, updating only the adapter weights. Additionally, we apply a random permutation to the dimensions of the LM embeddings before passing them to the adapter, encouraging the adapter to learn a mapping that is invariant to the ordering of embedding dimensions. During inference-time ensembling, each ensemble member uses a different permutation, and predictions are averaged. We initialize the adapter from the original TabPFN encoder weights. TabPFN's feature encoder processes groups of 3 features at a time, concatenating 3 NaN-mask indicators, giving a total input size of 6. Writing the encoder weight matrix as $W_{\text{enc}} \in \mathbb{R}^{d_{\text{out}} \times 6}$, we denote by $W_{\text{enc}}^{[:,3]}$ the first 3 columns corresponding to the feature inputs. We initialize the $i$-th text-cell projection as

$$W_{\text{cell}_i} = W_{\text{enc}}^{[:,3]} R_i, \tag{1}$$

where $R_i \in \mathbb{R}^{3 \times d_{\text{LM}}}$ is a random matrix with $R_{ij} \sim \mathcal{N}(0, 1)$ and $d_{\text{LM}}$ is the LM embedding size. Further training details and hyperparameters are provided in Appendix A.

---

[1] https://huggingface.co/sentence-transformers/all-MiniLM-L6-v2

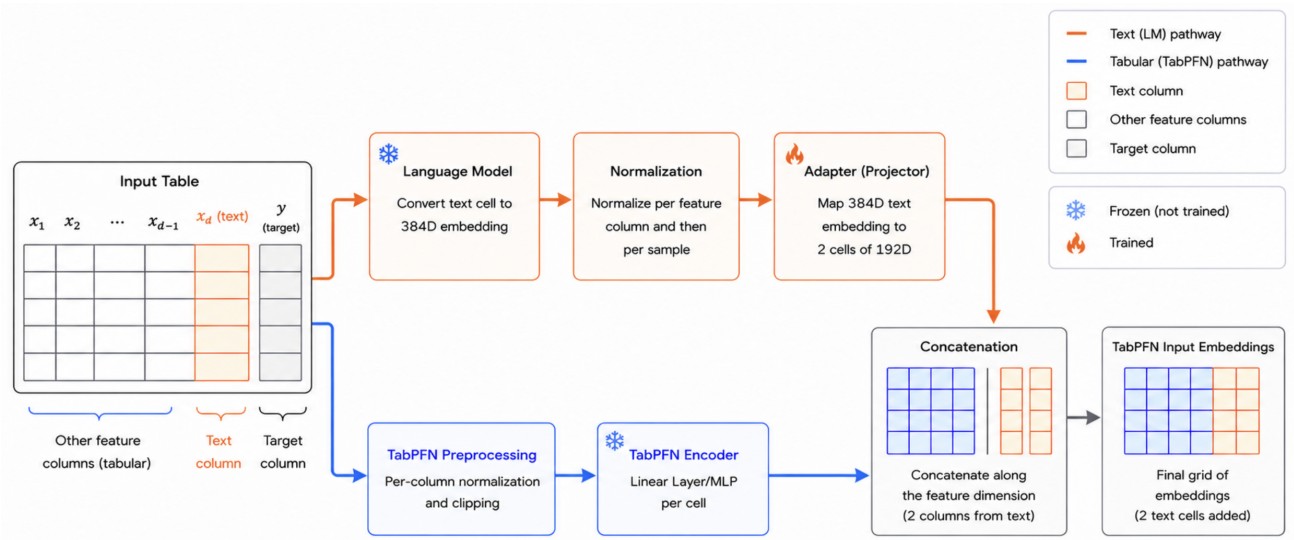

*Figure 1.* **Overview of our text adapter pipeline.**

## 4. Results

We evaluated our approach on TextTabBench (Mráz et al., 2025). TextTabBench contains 13 datasets (6 for classification and 7 for regression) curated specifically to test tabular pipelines on datasets with free long text columns. In contrast, the data collection we use for pretraining encompasses a wider variety of string types, including shorter strings and categorical-like text that are not full sentences.

We compare our approach against ConTextTab, which inherently handles textual features. To establish rigorous baselines for standard TabPFN, we include three variants: (1) **TabPFN (text dropped)**, which applies the default TabPFN after stripping all textual columns, serving as a lower bound to measure the informativeness of the text; (2) **TabPFN (tf-idf)**, which extracts standard term frequency-inverse document frequency features from the text. For this, we utilize the default `StringEncoder` from the `skrub` (skrub developers, 2024) library, which first applies a tf-idf vectorization and subsequently reduces the dimensionality via Truncated SVD; and (3) **TabPFN (PCA-30)**, a pipeline that embeds the text using a language model, normalizes across the respective text column, then applies PCA to reduce the embeddings to 30 dimensions and finally provides these values as numerical features to TabPFN. We use 30 dimensions since this is the default in skrub's `TextEncoder`. For regression, we disable the safepower preprocessing transformation of TabPFN across our adapter runs and baselines, as it decreased performance.

All evaluations were conducted using 3-fold cross-validation. For our adapter, we apply per-feature normalization followed by per-sample normalization on the sentence embedder's output embeddings. We trained separate adapters for classification and regression tasks. In its default setting, TabPFN, like many other tabular foundation models, will average across multiple forward passes with different types of preprocessing of the same data. During this inference-time ensembling, each ensemble member uses a different permutation of embedding dimensions before the adapter, and predictions are averaged across ensemble members. For classification, we report min-max normalized ROC-AUC, computed per dataset and fold as $(\mathrm{AUC}_{\mathrm{model}} - \mathrm{AUC}_{\mathrm{worst}})/(\mathrm{AUC}_{\mathrm{best}} - \mathrm{AUC}_{\mathrm{worst}})$, where the $\mathrm{AUC}_{\mathrm{worst}}$ and $\mathrm{AUC}_{\mathrm{best}}$ are the minimum and maximum scores across all evaluated models on each fold respectively. Results are then averaged across folds and datasets. For regression, we report min-max normalized RMSE according to the following formula: $(\mathrm{RMSE}_{\mathrm{model}} - \mathrm{RMSE}_{\mathrm{best}})/(\mathrm{RMSE}_{\mathrm{worst}} - \mathrm{RMSE}_{\mathrm{best}})$, using the same per-fold computation and averaging scheme.

Table 1 demonstrates the average results on different task types. For regression, our results show that the TabPFN Text Adapter outperforms all other methods while occupying, for example, far fewer tokens than the PCA-30 baseline; this provides a significant computational efficiency advantage due to the quadratic complexity of attention. For classification, however, while close, it is not on par with the PCA-30 baseline and is notably behind ConTextTab.

## 5. Ablations

In Table 2, all reported values are normalized relative to the default configuration. For classification, we report

$$\Delta \mathrm{AUC} = \frac{\mathrm{AUC}_{\mathrm{model}} - \mathrm{AUC}_{\mathrm{default}}}{\mathrm{AUC}_{\mathrm{default}}},$$

*Table 1.* **Average performance on TextTabBench.** Classification: min-max normalized ROC-AUC; Regression: max-min normalized RMSE. Best results in bold.

| Model | Classification (%) ↑ | Regression (%)↓ |
|---|---|---|
| ConTextTab | **97.65** | 32.9 |
| TabPFN (text dropped) | 0.0 | 81.2 |
| TabPFN (tf-idf) | 56.8 | 54.7 |
| TabPFN (PCA-30) | 86.4 | 38.9 |
| **TabPFN (ours)** | 84.5 | **23.3** |

*Table 2.* **Ablation study on TextTabBench.** Default settings marked (*).

| Type | Config | Classification ↑ | Regression ↑ |
|---|---|---|---|
| Default | | 0.0 | 0.0 |
| Token Count | 1 (REG*) | -1.17 | 0.0 |
| | 2 | -0.74 | -0.13 |
| | 5 (CLF*) | 0.0 | -0.77 |
| | 10 | 0.20 | -0.81 |
| Fusion Depth | Layer 1 | -2.00 | -3.33 |
| | Layer 2 | -1.86 | -2.60 |
| | Layer 3 | -4.26 | -2.84 |
| Language model | e5-small-v2 | -0.51 | 0.28 |
| Normalization | None | -2.73 | -3.28 |
| Initialization | Random | -0.63 | -3.64 |

and for regression, we report

$$\Delta\text{RMSE} = \frac{\text{RMSE}_{\text{default}} - \text{RMSE}_{\text{model}}}{\text{RMSE}_{\text{default}}}.$$

Therefore, the default row is always 0.00 by definition.

To thoroughly understand the components driving our adapter's performance, we conducted a series of ablations focusing on architectural choices and hyperparameters.

**Number of projection tokens.** we varied the number of tokens to which the adapter projects. While increasing the token count theoretically reduces the information bottleneck by allowing the language model to communicate richer semantic details to TabPFN, it also increases the computational complexity of the subsequent attention mechanisms. We find that the optimal token count differs between task types: ten tokens work best for classification, while a single token is sufficient—and superior—for regression. The difference between classification and regression could be due to fundamental differences in their pretraining as they were trained on different synthetic priors and also the slight differences in their architectures. For instance, the feature encoder differs: a linear layer for classification versus an MLP for regression.

**Fusion depth.** we varied the transformer layer at which the adapted text embeddings are concatenated with TabPFN's

native embeddings. By default, our adapter performs early fusion (prior to the first transformer layer). At this stage, a standard TabPFN token represents a simple group of normalized numerical features, which may limit the representational capacity expected of an incoming text token. Fusing at a deeper layer might allow a single token to represent more complex abstract concepts. However, this late fusion inherently reduces the number of joint attention layers applied to the concatenated sequence, potentially restricting TabPFN's ability to learn complex relational patterns between textual and numerical features in-context. Empirically, fusion at later layers consistently underperforms our default early-fusion setting.

Furthermore, we tested pretraining with a different language model (e5-small-v2) to assess the robustness of the pipeline to language model choice yielding very comparable results. Also, we evaluated the impact of removing our normalization strategies, which were added with the idea of a test-time dataset conditioning through the per-feature normalization and removing it leads to a significant performance drop. Finally, the effectiveness of our initialization strategy was tested, the point of using it was to have a strong starting point for the adapter that would output embeddings approximately aligned with TabPFN's expected embedding distribution. While for classification the effect is modest, it does clearly provide benefits in the case of regression.

## 6. Conclusion

In this paper, we introduced the TabPFN Text Adapter, a pretrained projector that maps LM embeddings directly into TabPFN's embedding space, enabling the state-of-the-art foundation model for tabular data to leverage textual features. We demonstrated that with our adapter, we were able to outperform all baselines for regression and, for classification, achieve very close performance to the PCA-30 baseline, which is the best-performing pipeline with TabPFN.

**Limitations and Future Work.** A key limitation of our adapter is that it applies the same pretrained linear layer or MLP across all tasks, but the combination of dimensions that are important varies between tasks. Ideally, TabPFN could use its statistical reasoning to determine what dimensions of the LM's high dimensional embedding are most important for the task at hand, but we assume that mapping into TabPFN's embedding space introduces significant information loss, since a single token inside TabPFN's embedding space likely represents much less information than what the LM's high dimensional embedding contains. Thus, a promising direction for further research is a version of our adapter, that can additionally condition across the full text column, or even across other columns in the table, in particular also the targets, in order to determine how to reduce the dimensions.

## Acknowledgement

Funded by the European Union. Views and opinions expressed are however those of the author(s) only and do not necessarily reflect those of the European Union or the European Commission. Neither the European Union nor the European Commission can be held responsible for them. This work was supported by the European Union's Horizon Europe research and innovation programme under grant agreement No 101214398 (ELLIOT).

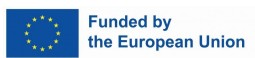 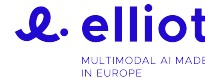

Lennart Purucker acknowledges funding by the Deutsche Forschungsgemeinschaft (DFG, German Research Foundation) under SFB 1597 (SmallData), grant number 499552394. Frank Hutter acknowledges the financial support of the Hector Foundation. Mustafa Tajjar is supported by the Konrad Zuse School of Excellence in Learning and Intelligent Systems (https://eliza.school/) through the DAAD programme Konrad Zuse Schools of Excellence in Artificial Intelligence, sponsored by the Federal Ministry of Education and Research. We thank the reviewers for their feedback.

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

## A. Pretraining Setup and Hyperparameters

We pretrain the adapter with AdamW using a learning rate of $2 \times 10^{-4}$, and gradient accumulation of 5 dataset batches. Gradient clipping is applied with max norm 2.0 at the dataset step level and 0.5 before each optimizer update. For TabPFN forwards during training, we use 2 estimators and fusion depth 0.

We use `all-MiniLM-L6-v2` as sentence encoder which has an output dimension of 384 and project text into one or several of TabPFN's 192-dimensional embedding space tokens.

For pretraining, for each dataset we randomly subsample between 1 and 2 text columns, we also drop a random fraction (0.0–0.3) of the other columns. Lastly we sample row counts from {1000, 2000}.

## B. Per-Dataset Results on TextTabBench

Tables 3 and 4 report per-dataset scores for all methods on the TextTabBench classification and regression tasks respectively. Best results per dataset are bolded.

*Table 3.* Per-dataset ROC-AUC on TextTabBench classification tasks (↑).

| Dataset | ConTextTab | TabPFN (text dropped) | TabPFN (tf-idf) | TabPFN (PCA-30) | TabPFN (ours) |
|---|---|---|---|---|---|
| Customer Complaints | **0.8712** | 0.7703 | 0.8211 | 0.8448 | 0.8470 |
| HS Cards | 0.8757 | 0.7853 | 0.8058 | **0.8819** | 0.8539 |
| Job Frauds | 0.9950 | 0.9177 | 0.9875 | **0.9954** | 0.9944 |
| Kickstarter | 0.8869 | 0.7877 | 0.8843 | **0.8946** | 0.8918 |
| OSHA Accidents | **0.6099** | 0.5899 | 0.6002 | 0.5991 | 0.6032 |
| Spotify | **0.9921** | 0.9571 | 0.9696 | 0.9908 | 0.9899 |

*Table 4.* Per-dataset RMSE on TextTabBench regression tasks, normalized by $\text{std}(y)$ (↓).

| Dataset | ConTextTab | TabPFN (text dropped) | TabPFN (tf-idf) | TabPFN (PCA-30) | TabPFN (ours) |
|---|---|---|---|---|---|
| AirBnB | 0.5681 | 0.5665 | 0.5659 | **0.5615** | 0.5631 |
| Beer | **0.5748** | 0.6603 | 0.6301 | 0.6037 | 0.5921 |
| Calif. Houses | 0.5733 | 0.5017 | 0.5036 | 0.5112 | **0.4735** |
| Laptops | **0.3088** | 0.3986 | 0.3605 | 0.3762 | 0.3616 |
| Mercari | 0.8454 | 0.9959 | 0.8949 | **0.7977** | 0.8474 |
| SF Permits | 0.6439 | 0.6933 | 0.6896 | 0.6694 | **0.6267** |
| Wine | **0.7042** | 0.7680 | 0.7444 | 0.7310 | 0.7294 |

## C. Pretraining Datasets

The datasets utilized during the pretraining phase are listed below, separated by their respective task types into classification (Table 5) and regression (Table 6). The STRABLE benchmark from which the datasets are chosen focuses on tabular datasets with strings. We used 15 classification and 35 regression tasks, while ensuring that the subset of STRABLE datasets we used does not have any overlap with TextTabBench.

*Table 5.* Regression datasets utilized during pretraining.

| Classification Datasets | |
| --- | --- |
| aca-federal-upper-limits-wide | china-overseas-finance-inventory |
| clear-corpus | college-deposit-product-marketing |
| community-banking_wide | conflict-events_wide |
| contract-awards-investment-project-financing | contributions-to-financial-intermediary-funds |
| external-clinician-dashboard | financial-intermediary-funds-funding-decisions |
| financial-management | foreign-gift-and-contract |
| fts-requirement-and-funding | gainful-employment |
| global-dams-database | historic-perimeters-wildfires |
| hospitals | ibrd-statement-loans-guarantees |
| ida-statement-credits-grants-guarantees | ifc-advisory-services-projects |
| industry-payments-project | insurance-company-complaints |
| journal-ranking_wide | local-government-renewable-action |
| local-law-enforcements | medically-underserved-areas-populations |
| mlr-summary-reports | national-average-drug-acquisition-cost |
| oil-natural-gas-platform | pol-terminal |
| power-plants | prison-boundaries |
| recipient-executed-grants-commitments-disbursements | summary-of-deposit_wide |
| us-school-bus-fleet | |

*Table 6.* Classification datasets utilized during pretraining.

| Regression Datasets | |
| --- | --- |
| animalandveterinary-event | chocolate-bar-ratings |
| device-covid19serology | drug-drugsfda |
| drug-ndc | food-enforcement |
| food-event | historical-earthquake-locations |
| historical-volcanic-locations | lending-club-loan |
| medicines | michelin-ratings |
| prepaid-financial-product | rasff_window |
| tobacco-problem | |

