# OpenReview forum: "Towards Pretraining Text Encoders for TabPFN"
_ICML.cc/2026/Workshop/FMSD — FMSD @ ICML 2026 Poster_

### Official Review · Reviewer_gY47 · 2026-05-17
**Useful lightweight text adapter for TabPFN, but evaluation is still limited**

**Rating:** 6
**Confidence:** 3

**Review:**

**Summary:**

The paper proposes a TabPFN Text Adapter, which maps frozen sentence-transformer embeddings into TabPFN’s internal embedding space. The goal is to avoid the PCA bottleneck commonly used when adding text features to TabPFN, while keeping both TabPFN and the language model frozen.

**Strengths:**

 The problem is important and well aligned with the workshop, since many real tabular datasets contain free-text columns. The proposed adapter is simple, efficient, and practically useful. Freezing both backbones makes the method much cheaper than end-to-end text-tabular pretraining. The regression results on TextTabBench are promising, and the ablations on token count, fusion depth, normalization, initialization, and language model choice are useful.

**Weaknesses:**

 The empirical evaluation is still limited. The method is tested only on TextTabBench, and the pretraining data is internally curated with limited detail, which affects reproducibility. The classification results are also not fully convincing: the adapter is slightly worse than the PCA-30 baseline and much worse than ConTextTab. The paper would benefit from more analysis of when the adapter helps, especially compared with simple PCA or TF-IDF pipelines. Runtime and memory savings are claimed but not quantified.

**Detailed comments:**

The authors should report wall-clock inference cost and token/attention cost compared with PCA-30 and ConTextTab. More details about the pretraining data would improve reproducibility. The title could be revised, since the method does not pretrain the text encoder itself; it trains an adapter on top of a frozen text encoder. It would also be useful to test stronger or larger sentence encoders and more datasets beyond TextTabBench.

**Justification:**

This is a relevant and practical workshop paper with a clean idea and promising regression results. However, the classification performance and limited evaluation make the contribution less convincing as a general solution for text-aware TabPFN.

---

### Official Review · Reviewer_3KJK · 2026-05-21

**Rating:** 6
**Confidence:** 3

**Review:**

Summary
- The authors introduce a pretrained text-to-TFM token projector, the TabPFN Text Adaptor, to improve performance on high-cardinality data by lightweight training that maps text embeddings into the target space. This design addresses the bottleneck of prior approaches based on PCA while improving efficiency.

Strengths
- Promising results, with gains on regression tasks and performance close to the PCA-30 baseline on classification tasks.
- Comprehensive presentation of the pipeline, methodology, and experimental setup.

Areas for Improvement
- As the authors mention, the embedding-space mapping should be further optimized to reduce information loss.
- The paper does not provide sufficient analysis or insights into how the proposed method could match or outperform TabPFN (PCA-30) or ConTextTab.

---

### Official Review · Reviewer_FxCa · 2026-05-22
**Review of "Towards Pretraining Text Encoders for TabPFN"**

**Rating:** 4
**Confidence:** 3

**Review:**

End-to-end pretraining often relies on low-quality, web-scraped tables that poorly represent actual predictive tasks. Forcing LLMs to jointly process all inputs disrupts the strong mathematical priors tabular models learned from synthetic data, degrading their performance on purely numerical and categorical features. Furthermore, representing simple scalars with high-dimensional dense vectors is computationally wasteful, introducing significant processing overhead and dimensional noise.